# Investigation of the Genomic and Pathogenic Features of the Potentially Zoonotic *Streptococcus parasuis*

**DOI:** 10.3390/pathogens10070834

**Published:** 2021-07-02

**Authors:** Jianping Wang, Xueli Yi, Pujun Liang, Yuanmeihui Tao, Yan Wang, Dong Jin, Bin Luo, Jing Yang, Han Zheng

**Affiliations:** 1State Key Laboratory of Infectious Disease Prevention and Control, National Institute for Communicable Disease Control and Prevention, Chinese Center for Disease Control and Prevention, Changping, Beijing 102206, China; wangjianping@icdc.cn (J.W.); liangpujun1997@163.com (P.L.); taoyuanmeihui@163.com (Y.T.); wangyan@icdc.cn (Y.W.); jindong@icdc.cn (D.J.); yangjing@icdc.cn (J.Y.); 2The Affiliated Hospital of Youjiang Medical University for Nationalities, Clinical College of Youjiang Medical University for Nationalities, Youjiang 533000, China; 1566@ymcn.edu.cn (X.Y.); 1558@ymcn.edu.cn (B.L.)

**Keywords:** *S. parasuis*, human infection, virulence, histopathologic changes, pro-inflammatory cytokines, antimicrobial resistance, phylogeny, *cps*

## Abstract

Recently, *Streptococcus suis* reference strains of serotype 20, 22, and 26 were reclassified as *Streptococcus parasuis*. The public health significance of *S. parasuis* is underestimated due to the lack of clinical isolates. In the present study, we first reported two sporadic *S. parasuis* infections in humans, after using full-length 16S rRNA and housekeeping genes’ phylogeny and ANI values of genome sequence comparisons to determine the species of their isolates BS26 and BS27. Compared to highly pathogenic *S. suis* strain P1/7, *S. parasuis* strains BS26 and BS27 possessed a delayed capacity to initiate lethal infection, which may attribute to the later production of higher level of pro-inflammatory cytokines. Differed to *S. suis* strain P1/7, *S. parasuis* strains did not induce significant inflammatory response in the brain of mice. Histopathological changes in liver and lungs were widely present in mice infected with *S. parasuis* strains. Our data indicated that the pathogenic mechanism of *S. parasuis* may be different from that of *S. suis*. Three lineages in the core-genome phylogenetic tree and ten types of *cps* gene cluster were found in 13 *S. parasuis* genomes, indicating high heterogeneity of this species. The similarity of CPS structure and antibiotic-resistant genes relative to *S. suis* indicated the evolutionary affinity between the two species. Our data suggested *S. parasuis* is a potential zoonotic pathogen and poses severe threat to health of susceptible people. Further study on the epidemiology and public health significance of *S. parasuis* is urgently necessary.

## 1. Introduction

*Streptococcus suis* is an important zoonotic pathogen that causes primarily meningitis, sepsis, endocarditis, arthritis, and pneumonia in both pigs and humans [1,2]. Currently, 35 serotypes (types 1 through 34 and 1/2) of *S. suis* have been identified on the basis of their capsular polysaccharide antigens (CPS) [1,3]. *S. suis* serotypes 32 and 34 have been reclassified as *Streptococcus orisratti* [4]. Recently, *S. suis* reference strains of serotypes 20, 22, 26 were proposed as *Streptococcus parasuis* [5] and serotype 33 was reclassified as *Streptococcus ruminantium* [6]. The presence of *S. parasuis* in diseased pigs and calves with pneumonia or systemic infection (meningitis, arthritis, endocarditis, or septicemia) indicated that *S. parasuis* may be pathogenic to pigs and/or calves [7,8,9,10]. The public health significance of *S. parasuis* has not been evaluated thoroughly, due to the lack of clinical isolates and published case reports. Strains BS26 and BS27 isolated from two patients and reported in the current study were classified as *S. parasuis* species by combining full-length 16S rRNA and housekeeping genes’ phylogeny and ANI values of genome sequence comparisons. To evaluate the potential virulence of BS26 and BS27, the survival curve, histopathological lesions, and kinetics of inducing pro-inflammatory cytokines production in vivo were compared to those of highly pathogenic *S. suis* strain P1/7 [11]. Additional 11 genomes of *S. parasuis* available in NCBI were included to investigate the phylogeny and genomic features of *S. parasuis*.

## 2. Results

### 2.1. Phylogenetic Analysis of Full-Length 16S rRNA and Housekeeping Genes groEL, gyrB, sodA, and recN of S. parasuis

Strain BS27 had 16S rRNA gene sequence similarity with BS26 equal to 100%. Phylogenetic analysis using 16S rRNA gene sequences demonstrated that strains BS26 and BS27 were 98.9% similar to *S. parasuis* type strain SUT-286^T^ but 96.6% similar to *S. suis* type strain NCTC10234^T^ (Appendix A). A dendrogram was constructed from a similarity matrix using the 16S rRNA sequences from 16 *S. parasuis* strains (the full-length 16S rRNA gene sequences were not extracted from draft genomes SUT-319 and SUT-380), *S. suis* type strain NCTC10234^T^, and *E. faecalis* JCM 5803 (Figure 1). *S. suis* type strain NCTC10234^T^ and *E. faecalis* JCM 5803 were distant from *S. parasuis* strains that fell into a group. Compared with other 14 *S. parasuis* strains, the similarity of 16S rRNA sequences of BS26 and BS27 ranged from 98.9% to 99.6% (Appendix A). The similarity of 16S rRNA sequences among 16 *S. parasuis* ranged from 98.8% to 100%. The similarity of 16S rRNA sequences between *S. parasuis* and *S. suis* ranged from 96.5% to 97.6%. (Appendix A). 

In addition to the 16S rRNA gene, housekeeping genes such as *groEL*, *gyrB*, *sodA*, and *recN* were also suitable and helpful for *Streptococcus* strain differentiation at the species level [12,13]. Strain BS27 shared 100% housekeeping-gene sequence similarity with BS26. Sequence analysis of housekeeping genes *groEL*, *gyrB*, *sodA*, and *recN* demonstrated that strains BS26 and BS27 were 97.6%, 97.6%, 97.4%, and 98.7% similar to *S. parasuis* type strain SUT-286^T^, respectively (Appendix A). Consistent with the 16S rRNA gene result, *S. parasuis* strains were grouped together and were well separated from *S. suis* NCTC10234^T^ in the phylogenetic trees that were based on the *groEL*, *gyrB*, *sodA*, and *recN* genes (Figure 1). Divergence values of 0–5.8%, 0–2.8%, 0–2.7%, and 0–3.1% among *S. parasuis* strains were obviously lower than 7–8.5%, 15.7–16.3%, 23–25%, and 15.3–17% with *S. suis* strain NCTC10234^T^ for the *groEL*, *gyrB*, *sodA*, and *recN* genes, respectively (Appendix A).

Strains BS26 and BS27 had an average nucleotide identity (ANI) value of 95.1% and 95.2% with *S. parasuis* type strain SUT-286^T^, respectively (Appendix A). In contrast to the high similarity with *S. parasuis* type strain SUT-286^T^, ANI values of BS26 and BS27 were 83.42% and 83.33% with *S. suis* type strain NCTC10234^T^, respectively. ANI values among 13 genomes of *S. parasuis* were in the range 93.8–99.99%, obviously higher than the range 83.33–84.72% with *S. suis* type strain NCTC10234^T^ (Appendix A).

### 2.2. Difference in Virulence between S. suis Strain P1/7 and Two S. parasuis Strains

In order to evaluate the potential virulence of *S. parasuis* BS26 and BS27, we compared the survival curve of C57BL/6 mice infected with two *S. parasuis* strains and highly pathogenic *S. suis* strain P1/7. Most mice infected with P1/7, BS26 or BS27 showed severe septic signs in period of the infection, such as rough hair coat, swollen eyes, weakness, and shivering. 

The survival rate of mice infected with *S. suis* strain P1/7 or two *S. parasuis* strains was significantly different from that of mock-infected mice. Moreover, significant difference in the survival level was also observed between mice infected with *S. suis* strain P1/7 and two *S. parasuis* strains, which attributed to the differences at an early phase of the infection. Mice infected with *S. suis* strain P1/7 had a 20% survival rate at 8 h post-infection, while mice infected with *S. parasuis* strain BS26 and BS27 had a 95% and 90% survival rate at the same time point, respectively. It is noteworthy that the survival levels of mice infected with BS26 and BS27 dramatically decreased after 12 h post-infection. At 24 h post- infection, survival rates of the P1/7-infected group (15%), BS26-infected group (20%), and BS27-infected group (25%) were very similar (Figure 2 and Appendix A). 

### 2.3. Histopathological Lesions and Bacterial Load in Survival Mice

Totally, three mice infected with *S. suis* strain P1/7, three mice infected with *S. parasuis* strain BS26, and five mice infected with *S. parasuis* strain BS27 survived at 72 h post-infection in survival experiment. Strains were isolated in the peripheral blood, brain, lung, liver, spleen, and kidney of all survival mice. 

Slight deformation of neuron was found in the brains of seven mice infected with *S. parasuis* strain (Figure 3A(1)). Only one mouse infected with *S. parasuis* BS26 was observed slightly infiltrated by lymphocytes (Figure 3A(1)). Histopathological lesions of *S. parasuis*-infected mice were mainly observed in the lung and liver. These organs presented bacterial loads over 10^2^ CFU/0.1 g. Hepatocyte steatosis was significant histopathological changes of livers and was present in all *S. parasuis*-infected mice (Figure 3B(1)). Coagulative necrosis was observed in the liver of two mice infected with the *S. parasuis* strain (Figure 3B(1)). Alveoli wall thickening with diffused infiltration of neutrophils and lymphocytes was observed in the lung of all *S. parasuis*-infected mice (Figure 3C(1)). Some hemorrhagic foci in lungs were also found in the two mice infected with *S. parasuis* strain (Figure 3C(2)). No significantly histopathological changes were observed in the heart and spleen of *S. parasuis*-infected mice. 

Different from *S. parasuis* strains-infected mice, neuronal necrolysis and significant infiltration by neutrophils were observed in brains of all mice infected with *S. suis* strain P1/7 (Figure 3A(2)). Although bacterial loads ranged from 10^2^ to 10^4^ CFU/0.1 g in other organs of mice infected with *S. suis* strain P1/7, no significantly histopathological lesions were observed in them (Figure 3B(2),C(3)). Bacterial counts of mice infected with *S. suis* were significantly higher than those of mice infected with *S. parasuis,* except for the liver (Figure 3D).

### 2.4. Pro-Inflammatory Cytokine Production and Bacterial Loads in Mice Infected with S. parasuis Strain BS26 and S. parasuis Strain P1/7

Proinflammatory cytokines were responsible for the death of mice infected with *S. suis* strains within 24 h post infection [14,15,16]. The kinetics of proinflammatory cytokines induced by P1/7 and BS26 within 24 h post-infection was compared. Stimulations with *S. parasuis* strain BS26 and *S. suis* strain P1/7 induced time-dependent production of IL-6 and TNF-a in mice. The kinetics and levels of IL-6 and TNF-a that were induced in vivo were different between the two strains. *S. suis* strain P1/7 was able to induce significantly more rapid production and higher levels of IL-6 and TNF-a early in the post-infection period, reaching peaks of 54,027 pg/mL and 2067 pg/mL at 8 h, respectively. Then, levels of cytokines gradually decreased from 12 h to 24 h post-infection. The pattern of TNF-a induced by BS26 in vivo was similar to those patterns of cytokines induced by P1/7, peaking at 8 h post-infection and gradually decreasing thereafter. In contrast, the level of IL-6 induced in vivo by BS26 peaked at 34,690 pg/mL at 12 h post-infection and gradually decreased thereafter. Moreover, BS26 induced higher levels of IL-6 than P1/7 did at 12 h post-infection. Significantly higher levels of cytokines were produced by *S. suis* strain P1/7 at 4, 8, 16, and 24 h post-infection than those produced by *S. parasuis* strain BS26 (Figure 4A,B).

In the present study, we also counted the bacterial loads in the peripheral blood, liver, lung, and brain of infected mice at 4, 8, 12, 16, and 24 h post-infection. The bacterial counts in the peripheral blood and organs of mice infected with *S. suis* strain P1/7 were statistically higher than those of *S. parasuis* strain BS26-infected mice. 

In *S. parasuis* strain BS26-infected group, bacterial counts in the peripheral blood reached 10^6^ CFU/mL at 4 h post-infection; and then descended dramatically to 10^4^ CFU/mL at 8 and 12 h post-infection, and 10^2^ CFU/mL at 24 h post-infection (Figure 4C). Viable counts in the brain presented similar kinetics, peaked at 10^4^ CFU/0.1 g at 4 h post-infection and gradually decreased to 10^3^ CFU/0.1 g at 24 h post-infection (Figure 4D). In contrast, bacterial counts in the liver and lung showed a different pattern. Viable counts in the liver and lung reached approximately 10^6^ CFU/mL at 4 h post-infection, peaked over 10^6^ CFU/0.1 g at 12 h post-infection and gradually decreased to 10^5^ CFU/0.1 g at 24 h post-infection (Figure 4E,F). It is noteworthy that bacterial counts in the liver and lung were higher than that of peripheral blood from 8 h to 24 h post-infection. 

In *S. suis* strain P1/7-infected group, bacterial counts in the peripheral blood reached 10^8^ CFU/mL at 4 h post-infection, remained at 10^8^ CFU/mL at 8 and 12 h post-infection, and slightly decreased to 10^6^ CFU/mL at 24 h post-infection (Figure 4C). Kinetics of viable counts in the organs were similar to those of mice infected with *S. parasuis* strain BS26. Viable counts in the brain peaked at 10^8^ CFU/mL at 4 h post-infection and gradually decreased to 10^6^ CFU/mL at 24 h post-infection (Figure 4D). Viable counts in the liver and lung reached 10^6^ CFU/0.1 g at 4 h post-infection, peaked at 10^7^ CFU/0.1 g at 12 h post-infection, and slightly decreased to 10^6^ CFU/0.1 g at 24 h post-infection (Figure 4E,F).

### 2.5. AR Genes and Antimicrobial Susceptibility Profiles of BS26 and BS27

Only *msr(D)* and *mef(A)* genes that code for resistance to macrolides were found in genomes of BS26 and BS27. Lower levels of resistance to erythromycin and azithromycin was found in *S. parasuis* strains BS26 and BS27, and those levels were attributed to *msr(D)* and *mef(A)* genes with MICs 8 and 24 μg/mL, respectively. It is noteworthy that *S. parasuis* strains BS26 and BS27 were resistant to trimethoprim–sulfamethoxazole with an MIC > 32 μg/mL, although they did not harbor AR genes that were known to be related. In addition, *S. parasuis* strain BS26 and BS27 were susceptible to penicillin G, cefaclor, clindamycin, vancomycin, chloramphenicol, tetracycline, streptomycin, kanamycin, spectinomycin, and gentamicin.

### 2.6. Comparative Genomic Analysis of S. parasuis

The core-genome phylogenetic tree based on 442 core genes of 14 *Streptococcus* genomes indicated great diversity among *S. parasuis* genomes, which were clustered into three discrete lineages (Figure 5). Lineage 1 contained BS26, BS27, and four additional public genomes. Lineage 2 contained the reference genomes of serotype 20, 22, and 26, type strain SUT-286^T^, and two additional public genomes. Lineage 3 only contained SUT-7, which was distant from other *S. parasuis* genomes.

### 2.7. The Analysis of Cps Gene Clusters of S. parasuis

Complete *cps* gene clusters were extracted from eleven of 13 *S. parasuis* genomes. The three sides of *cps* gene clusters was deleted in genomes SUT-286^T^ and 4253. The *cps* gene cluster of BS26 was identical to that of BS27. The *cps* gene clusters of *S. parasuis* were located between the CDS1210 and CDS1209 genes (named as homologous genes in *S. suis* genome P1/7) and belonged to pattern III, identical to that of reference strains of serotype 20, 22, and 26 [17]. The fact that all *cps* gene clusters of *S. parasuis* harbored *wzy/wzx* genes indicated that CPs of *S. parasuis* were synthesized by the WZX/WZY pathway. The conserved *cpsA*, *cpsB*, *cpsC*, and *cpsD* were present and located on the 5′ side of the *cps* gene clusters. The central regions of these clusters were highly variable. The *cps* gene clusters were clustered into ten types (I to X) based on the sequence of the *wzy* gene (Table 1). Among them, *cps* types VI and X contained three genomes (SUT-319, SUT-328, and SUT-380) and two genomes (BS26 and BS27), respectively. Other *cps* types comprised one genome each. The *wzy* genes of type V, VII, IX, and X were highly homologous to those of *S. thermophilus* EPS type VI, *S. suis* serotype 24, *S. suis* serotype 10, and *S. thermophilus* EPS type V, respectively (Figure 6).

## 3. Discussion

Two strains—BS26 and BS27—were isolated from the blood cultures of two patients and initially identified as *S. suis* by MALDI-TOF MS. In the present study, the 16S rRNA gene sequence similarity between two strains and *S. parasuis* type strain SUT-286^T^ exceeded 98.7%, the threshold value to delineate the species [18]. Sequence analysis of their *groEL*, *gyrB*, *rpoB*, *sodA*, and *recN* genes also showed that the two strains had phylogenetic affinity with the *S. parasuis* species. 

ANI value can be used as a parameter to determine whether two genomes belong to the same species. The ANI cut-off value for species definition is species-dependent. Average ANI of the same species varies from 0.779 (*Polynucleobacter necessaries*) to 0.998 (*Leptospira biflexa*) [19]. By comparison of pairs of 1226 bacterial strains with whole genome sequences, bacterial species definition using cut-off value of 0.92 matched the current bacterial species definition well [19]. In the present study, BS26 and BS27 had ANI values of 95.1% and 95.2% with *S. parasuis* type strain SUT-286^T^, respectively. These ANI values are higher than the 95% cut-off ANI value for bacterial species that Goris et al. proposed [20]. It is important that full-length 16S rRNA and housekeeping genes’ and core-genome phylogeny demonstrated that *S. suis* type strain NCTC10234^T^ formed a distinct branch and was well separated from *S. parasuis*. By combining full-length 16S rRNA and housekeeping genes’ phylogeny and ANI values of genome sequence comparisons, it is concluded that the BS26 and BS27 strains belonged to the *S. parasuis* species. Before the present study, *S. parasuis* had been isolated from healthy and diseased pigs or calves [7,8,9,10]. The present study makes the first report of sporadic *S. parasuis* infections in humans. It is possible that the prevalence and clinical relevance of *S. parasuis* have been underestimated because of a lack of effective methods to identify it. 

To evaluate the potential virulence of *S. parasuis* strains BS26 and BS27, the survival curve, histopathological lesions, and kinetics of pro-inflammatory cytokines production in vivo were compared to those of highly pathogenic *S. suis* strain P1/7. Compared to *S. parasuis* strain BS26- and BS27-infected mice, those infected with *S. suis* strain P1/7 quite rapidly succumbed to the infection. Significant differences at the early phase of infection and similarity at the middle phase of infection in survival level were observed between *S. suis* strain P1/7-infected group and *S. parasuis* strains-infected groups. The results indicated that *S. parasuis* strains BS26 and BS27 possessed a delayed capacity to initiate lethal infection. 

The histopathological lesions in major organs were also investigated in all survival mice at 72 h post infection of survival assay. Consistent with the results of previous study [15], survival mice infected with *S. suis* strain P1/7 initiated significant inflammation in brains. Obviously higher bacterial loads present in brains of mice infected with *S. suis* P1/7 may contribute to the histopathological changes. No significant inflammatory response was found in the brains of the survival mice infected with *S. parasuis* strains. Histopathological lesions of survival mice infected with *S. parasuis* were widely present in lungs and livers. Hepatocyte steatosis was present in all *S. parasuis* infected mice. Hepatic mitochondrial and its enzyme activity may be impaired in response to sepsis induced by *S. parasuis,* causing hepatic lipid utilization disorder. Significant inflammation in lungs were observed in all survival mice infected with the *S. parasuis* strain. It is noteworthy that *S. parasuis* strain BS27 also led to the inflammation of both lungs in its host. 

In previous studies, no significant histopathological changes in organs were observed in dead mice infected with *S. suis* within 48 h post-infection [14,15]. Proinflammatory cytokines played a critical role in the death of mice or patients infected with *S. suis* within 24 h post-infection [15,16,21]. The overproduction of pro-inflammatory cytokines such as TNF-a and IL-6 plays an important role in the infection caused by *S. suis* [21]. IL-6 is an important inducer of acute phase proteins [22]. High levels of IL-6 correlate inversely with survival time in patients with sepsis [23]. TNF-α is one of the most important host mediators in the pathogenesis of septic shock [24] and may be responsible for the streptococcal toxic shock-like syndrome (STSLS) observed in the Sichuan outbreak [25]. In order to explicate the difference in mortality between mice infected with *S. suis* strain P1/7 and *S. parasuis* strains at early phase of infection, we also compared their kinetics of inducing TNF-a and IL-6 production in vivo within 24 h post-infection. The capacity of *S. parasuis* strain BS26 to induce TNF-α production in vivo is obviously lower than that of *S. suis* strain P1/7 during the whole infection. However, *S. parasuis* BS26 induced higher levels of IL-6 at 12 h post-infection. Interestingly, the survival levels of mice infected with *S. parasuis* srains dramatically decreased after 12 h post-infection. The later inflammatory response might result in most mice infected with *S. parasuis* strains succumbed to the infection at middle stage of survival experiment.

It is noteworthy that the bacterial loads of *S. parasuis* strain BS26 in peripheral blood and major organs were obviously lower than those of *S. suis* strain P1/7 during the infection. Moreover, *S. parasuis* strains BS26 and BS27 did not harbor “classical” *S. suis* virulence markers *mrp*, *sly*, and *epf* (data were not shown). Combined with the result of histopathology, we suspected the pathogenic mechanism of *S. parasuis* may be different from that of *S. suis*. 

In the present study, both *S. parasuis* strains carried AR genes *msr(D)* and *mef(A)* that are mainly associated with resistance to macrolide. The category and number of AR genes carried by *S. parasuis* strains BS26 and BS27 were lower than those of *S. suis* clinical strains [26,27]. The acquisition and dissemination of AR genes in *S. suis* are strongly associated with mobile genetic elements (MGEs), which are mainly integrative and conjugative elements (ICEs) and prophages [28]. We did not find MGEs in 13 genomes of *S. parasuis* (data were not shown). Further studies are needed to investigate the formation and dissemination mechanisms of AR genes in *S. parasuis* population.

In the present study, the phylogeny of 13 *S. parasuis* genomes were analyzed. The ANI values range from 93.8% to 99.99% indicating obvious heterogeneity among *S. parasuis* genomes. Consistent with the ANI values, the core-genome phylogeny clustered 13 *S. parasuis* genomes into three discrete lineages. *S. parasuis* strains BS26 and BS27 had phylogenic affinity with *S. parasuis* strain 4253 and were clustered into Lineage 1, which also contained three other genomes: SUT-319, SUT-328, and SUT-380. 

In the present study, the genetic characteristics of *cps* loci in each genome were investigated. As with that of reference strains of serotypes 20, 22, and 26, all complete *cps* gene clusters of *S. parasuis* belonged to pattern III [17]. They shared highly conserved *cpsA*, *cpsB*, *cpsC*, and *cpsD* genes and flanked regions with serotype reference strains of *S. suis*, indicating the phylogenetic affinity between *S. suis* and *S. parasuis*. The possession of *wzy/wzx* genes in each *cps* gene cluster strongly suggests that the CPs of all *S. parasuis* were synthesized using the WZX/WZY pathway. The *cps* gene clusters were clustered into ten groups on the basis of the sequences of *wzy* gene. In a previous study, SUT-286^T^ and SUT-380 agglutinated with the antisera of serotypes 20 and 22, respectively, whereas isolates SUT-7, SUT-319, and SUT-328 cross-reacted with antisera 22/26, 20/22, and 20/22, respectively [5]. However, obvious differences in sequences of the central regions including *wzx* and *wzy* genes were found among them in the present study. We reasonably speculate that the agglutination reactions were non-specific. Moreover, the central regions including *wzx* and *wzy* genes of four *cps* types of *S. parasuis* had high similarity with those of *S. thermophilus* and *S. suis,* respectively. Previous studies have shown recombination between *cps* gene clusters of *S. suis* and *S. pneumonia* [29,30]. These data indicated that intraspecies horizontal transfer of *cps* gene clusters among *Streptococcus* had frequently occurred. The presence of many different types of *cps* loci in *S. parasuis* also indicated high heterogeneity of this species. 

In conclusion, we first reported the sporadic *S. parasuis* infections in humans. Two *S. parasuis* strains isolated in patients possessed a delayed capacity to initiate lethal infection. Our data suggested *S. parasuis* as a potentially zoonotic pathogen could severely threat to health of susceptible people. Obvious heterogeneity was found among *S. parasuis* genomes. The similarity of CPS structure and AR genes relative to *S. suis* indicated the evolutionary affinity between the two species. Further study on the epidemiology and public health significance of *S. parasuis* is urgently necessary.

## 4. Materials and Methods

### 4.1. Bacterial Strains and Case Description

Strains BS26 and BS27 were isolated from blood cultures of peritonitis patient and arthritis patient with pneumonia in 2018, respectively. They were identified as *S. suis* by matrix-assisted laser desorption/ionization time-of-flight (MALDI-TOF, Microflex LRF, Bruker) method with log-score 1.82 and 1.85 (1.700 < score < 1.999 indicates “probable genus identification”), respectively. The two strains were confirmed as not belonging to *S. suis* by amplifying a nearly complete 16S rRNA gene using primer 27F(5′-AGAGTTTGATCMTGGCTCAG-3″) and 1492R (5′-TACGGYTACCTTGTTACGACTT-3′) and *S. suis*–specific *recN* and *gdh* genes [31,32,33,34]. Data of 16S rRNA sequencing and the PCR method detecting *recN* gene specific to *S. parasuis* [6] indicated that the two strains belonged to the *S. parasuis* species. 

On 9 June 2018, a 27-year-old female patient as host of BS26 was admitted to the Affiliated Hospital of Youjiang Medical University for Nationalities in Baise city because of peritonitis characterized by acute onset of fever (body temperature of 38.1 °C) and persistent periumbilical colic. Two years earlier, the patient was diagnosed with renal hypertension. The serum level of high-sensitivity C-reactive protein (hsCRP) (cobas c 702, Roche) measured by nephelometry and total counts of white blood cells (WBC) (Automated Hematology Analyzer XN series XN-20, sysmex) measured by flow cytometry were 31.11 mg/L and 10.2 × 10^9^/L, respectively. The patient’s blood pressure was 164/101 mm Hg. Ceftazidime and cefazolin sodium pentahydrate were introduced by antibiotic therapy. The patient recovered and was discharged. 

On 8 October 2018, a 53-year-old male patient as host of BS27 was admitted to the same hospital (as was the host of BS26) because of the acute onset of fever (body temperature of 38.9 °C), chill, cough, expectoration, and persistent pain in the right knee joint. His previous health status was well. The patient had been exposed to pigs one week prior to the onset of illness. A CT scan image indicated inflammation of both lungs. The serum level of hsCRP (cobas c 702, Roche) measured by nephelometry and total counts of WBC (Automated Hematology Analyzer XN series XN-20, sysmex) measured by flow cytometry were 54.84 mg/L and 22 × 10^9^/L, respectively. Cefatriaxone was introduced for two weeks. The patient recovered.

### 4.2. DNA Extraction, Sequencing, and Bioinformatic Analysis 

The strains were grown overnight on Columbia blood base agar (Thermo Fisher Scientific, Beijing, China) at 37 °C with 5% CO_2_. The genomic DNA of two strains was extracted and purified with the Wizard Genomic DNA Purification kit (Promega, Madison, WI, USA). A complete genome of BS26 was sequenced using PacBio Sequel platform and Illumina NovaSeq PE150. A draft genome of BS27 was sequenced using Illumina NovaSeq PE150. PacBio Sequel platform Libraries for single-molecule real-time (SMRT) sequencing was constructed with an insert size of 10 kb using the SMRT bell TM Template kit, version 1.0. In summary, the process was to fragment and concentrate DNA, repair DNA damage and ends, prepare blunt ligation reaction, purify SMRTbell Templates with 0.45X AMPure PB Beads, select size by using the BluePippin System, and repair DNA damage after size selection. Last, the library quality was assessed on the Qubit^®^ 2.0 Fluorometer (Thermo Fisher Scientific, Waltham, MA, USA), and the insert-fragment size was detected by Agilent 2100 (Agilent Technologies, Santa Clara, CA, USA). Sequencing libraries of Illumina NovaSeq platform were generated using NEBNext^®^ Ultra™ DNA Library Prep Kit for Illumina (NEB, Ipswich, England), by following the manufacturer’s recommendations; and index codes were added to attribute sequences to each sample. Briefly, the DNA sample was fragmented by sonication to a size of 350 bp, and then DNA fragments were end-polished, A-tailed, and ligated with the full-length adaptor for Illumina sequencing with further PCR amplification. Last, PCR products were purified using AMPure XP system (Beckman Coulter, Brea, CA, USA), and libraries were analyzed for size distribution by Agilent2100 (Agilent Technologies) and quantified using real-time PCR. The generated reads were assembled using SOAP*denovo* (release 1.04). Genes were predicted by using Glimmer 3.02, and gene orthologs were determined by using OrthoMCL V1.4. The complete genome of strain BS26 (1,932,292 bp) contained 2022 genes and had a G+C content of 39.75%. The draft genome of strain BS27 (1,909,795 bp) contained 2002 genes and had a G+C content of 39.69%. The coverage/depth of BS26 and BS27 genomes were 100%/288 and 98.84%/301, respectively. For comparison, 11 draft genomes of *S. parasuis* strains were also added, including serotype 20 reference strain 86-5192, serotype 22 reference strain 88-1861, serotype 26 reference strain 89-4109-1, SUT-7, SUT-286, SUT-319, SUT-328, SUT-380, 2674, 4253 [35], and 10-36905 (Table 1).

### 4.3. Phylogenetic Analysis of 16S RNA and Housekeeping Genes 

Full-length 16S rRNA, housekeeping genes *groEL*, *gyrB*, *sodA*, and *recN* sequences of the *S. parasuis* were retrieved from the GenBank database or assembled genomes and aligned by the program CLUSTAL W. With the MEGA 7 software (available online: www.megasoftware.net, accessed on 27 February 2021), the 16S rRNA and housekeeping genes’ sequence-based phylogenetic trees were constructed with neighbor-joining algorithms. The stability of the groupings was estimated by bootstrap analysis at the level of 1000 replications, and the genetic distances were calculated by the Kimura 2-parameter method. The corresponding sequences of *S. suis* type strain NCTC10234^T^ (Accession no. PRJEB6403) were included. The corresponding sequences of *Enterococcus faecalis* JCM 5803 (Accession no. NR_040789.1) or *S. suis* type strain NCTC10234^T^ were used as out-group to root the trees.

### 4.4. Experimental Infection

#### 4.4.1. Survival Assay

For comparison, a highly pathogenic S. suis strain P1/7 that was isolated from a pig with meningitis [11] was included in the present study. C57BL/6 mice (6 weeks old, female) were injected intraperitoneally with 5 × 10^7^ CFU of the live S. parasuis strain BS26, S. parasuis strain BS27, and S. suis strain P1/7 in 1 mL PBS or 1 mL PBS only as control group. Each strain-infected group contained ten mice and the mock-infected group contained five mice. The mortality was recorded per two hours within 12 h post-infection and per six hours from 12 h to 72 h post-infection. The experiment was performed independently in duplicate. Overall survival rates of the two experiments for each infected group were calculated via the Kaplan–Meier method.

#### 4.4.2. Histopathological Analysis and Bacterial Loads in Survival Mice

In survival assay, the peripheral blood, brain, lung, liver, spleen, and kidney of all survival mice at 72 h post-infection (three mice infected with S. parasuis strain BS26, five mice infected with S. parasuis strain BS27 and three mice infected with S. suis strain P1/7) were collected aseptically. 

One hundred microliters of serial tenfold dilutions of peripheral blood of each infected mouse were plated onto blood agar plates. Part of organs were accurately weighted and thoroughly ground to homogenate, placed in 1 mL of PBS. One hundred microliters of serial tenfold dilutions of organ homogenate were plated onto blood agar plates. Colonies were expressed as CFU/0.1 g for organ samples or CFU/mL for blood samples. The median values were used to express the bacterial counts of each group.

Another part of organs was fixed for 24 h at room temperature in 4% buffered formalin. After paraffin embedding, tissue sections conventional dewaxing to water were stained with H&E and dehydrated according to standard protocol. The images were acquired and analyzed under light microscopy.

#### 4.4.3. In Vivo Cytokine Production and Measurement of Bacterial Loads

C57BL/6 mice (6 weeks old, female) were injected intraperitoneally with 1 × 10^7^ CFU of the live *S. parasuis* strain BS26 and *S. suis* strain P1/7 strains in 1 mL PBS. Each group contained five mice. At 4 h, 8 h, 12 h, 16 h, and 24 h post-infection, the mice were killed, and the peripheral blood of each infected mouse was collected aseptically. These experiments were performed twice. All serum samples were tested with concentrations of IL-6 and TNF-α using the ELISA kit (R&D Systems, Minneapolis, MN, USA), as recommended by the manufacturer. The cytokine values were expressed as the median pg/mL values. The bacterial counts in peripheral blood, liver, lung, and brain of infected mice were measured according to the methods described in 4.4.2. The median values were used to express the bacterial counts of each group.

### 4.5. Detection of Antibiotic Resistance (AR) Determinants and Antimicrobial Susceptibility Profiles

We analyzed AR genes by searching the Comprehensive Antibiotic Resistance database (CARD) and the Antibiotic Resistance genes database (ARDB). We regarded a resistance gene as a homolog in the tested strains only if it exhibited at least 80% identity in the protein’s sequence across 80% of the protein’s length [36]. We tested for antimicrobial susceptibility by assessing the minimum inhibitory concentration (MIC) for all isolates by using an MIC-test strip (Liofilchem, Roseto degli Abruzzi, Italy). Those strips had a range of concentrations of penicillin G (0.002–32 μg/mL), cefaclor (0.016–256 μg/mL), tetracycline (0.016–256 μg/mL), erythromycin (0.016–256 μg/mL), azithromycin (0.016–256 μg/mL), clindamycin (0.016–256 μg/mL), chloramphenicol (0.016–256 μg/mL), vancomycin (0.016–256 μg/mL), streptomycin (0.064–1024 μg/mL), kanamycin (0.016–256 μg/mL), spectinomycin (0.064–1024 μg/mL), gentamicin (0.016–256 μg/mL), and trimethoprim-sulfamethoxazole (0.002–32 μg/mL). For quality control, we used *S. pneumoniae* ATCC 49619. For penicillin G, tetracycline, azithromycin, erythromycin, clindamycin, vancomycin, and chloramphenicol, breakpoints were used as recommended by the Clinical and Laboratory Standard Institute (CLSI) guidelines 2019 (M100-S29) for *Streptococcus spp. Viridans* group. For cefaclor and trimethoprim-sulfamethoxazole, breakpoints were used as recommended by the Clinical and Laboratory Standard Institute (CLSI) guidelines 2019 (M100-S29) for *S. pneumoniae*. No breakpoint values of streptomycin, kanamycin, gentamicin, and spectinomycin were available for *Streptococcus*. Their breakpoints were taken from previous studies of *S. suis* [37,38]. Breakpoints of antimicrobial agents are as follows: penicillin G, >4 μg/mL; cefaclor, >4 μg/mL; tetracycline, >8 μg/mL; azithromycin, >2 μg/mL; erythromycin, >1 μg/mL; clindamycin, >1 μg/mL; chloramphenicol, >16 μg/mL; vancomycin, >2 μg/mL; trimethoprim-sulfamethoxazole, >4 μg/mL; streptomycin, >250 μg/mL; kanamycin, >250 μg/mL; gentamicin, >250 μg/mL, and spectinomycin, >250 μg/mL.

### 4.6. Comparative Genomic Analysis of S. parasuis 

Thirteen *S. parasuis* genomes and genome of *S. suis* type strain NCTC10234^T^ were included in the comparative genomic analysis. The average nucleotide identity (ANI) analysis was performed on the OrthoANIu platform (available online: https://www.ezbiocloud.net/tools/orthoaniu, accessed on 20 November 2020) to define species boundaries of bacteria [39]. 

The whole genome phylogenetic tree based on 442 core genes of 14 genomes was constructed by Roary [40]. The core genes were extracted by BLASTP (96% identity, 99% of isolates a gene must be in to be core). The sequences of core genome were concatenated and aligned by MAFFT multiple sequence alignment program. The phylogenetic tree based on core genome was constructed by using the maximum likelihood algorithms in FastTree. The *S. suis* type strain NCTC10234^T^ was used as out-group to root the tree.

### 4.7. Analysis of cps Loci

Each *cps* locus sequence was extracted from the genomes. The TMHMM v2.0 analysis program (available online: http://www.cbs.dtu.dk/services/TMHMM/, accessed on 15 October 2020) was used to identify putative transmembrane proteins WZX and WZY. The *wzy* genes having a global match region at <50% of the amino-acid sequence and with an identity of <50% were identified as different *cps* types in pairwise comparisons between *S. parasuis* genomes. The sequence comparison was performed using blastN program in BLAST with an e-value cutoff of e-10 and was visualized using an in-house Perl script (available online: https://github.com/dupengcheng/BlastViewer, accessed on 2 March 2021). 

### 4.8. Statistics

The survival rates of different groups were compared using Log-rank test. Significant difference in bacterial counts between *S. suis* and *S. parasuis*-infected group was determined by Wilcoxon’s two-sample test. Statistical analysis of the cytokine data was performed by using the Wilcoxon’s two-sample test. For these tests, a *p*-value < 0.05 was considered to be significant.

### 4.9. Nucleotide Sequence Accession Numbers

The sequences of the 2 *S. parasuis* strains that were sequenced in the study were deposited in the GenBank under accession numbers CP069079 (BS26) and JAETXU000000000 (BS27).

## Figures and Tables

**Figure 1 pathogens-10-00834-f001:**
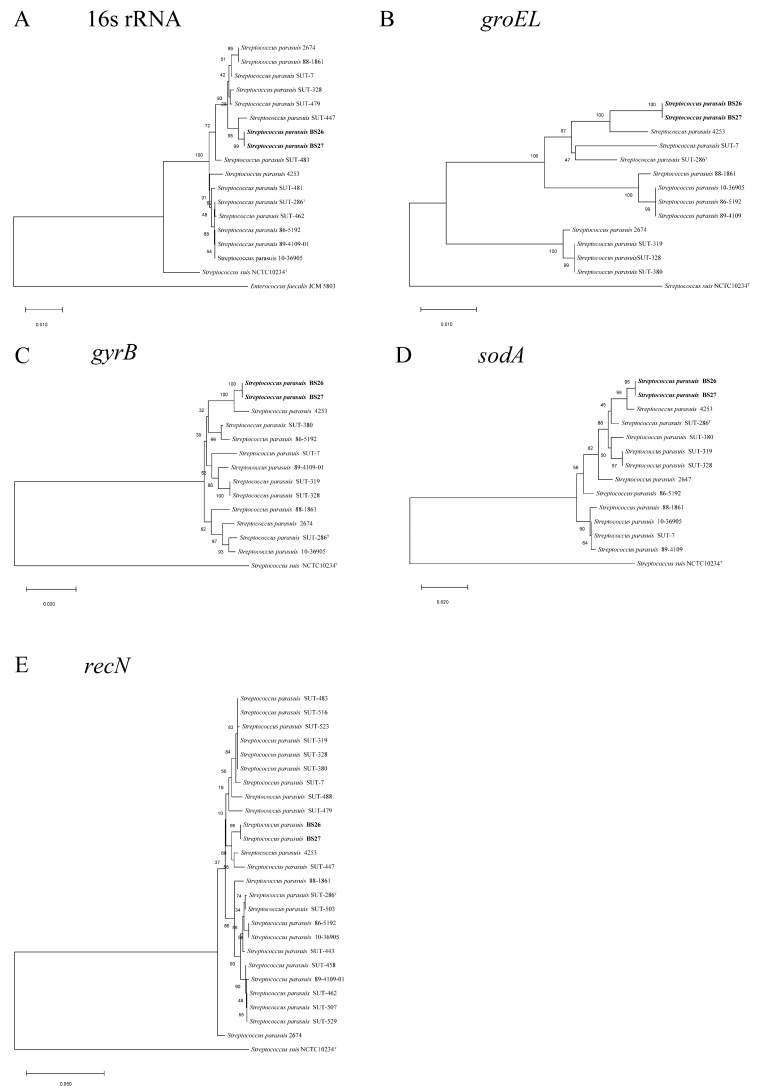
Phylogenetic relationships of strains used in the present study. The trees were based on an alignment of full length of the 16S rRNA (**A**), *groEL* (**B**), *gyrB* (**C**), *sodA* (**D**), and *recN* (**E**) by using the neighbor-joining method. The corresponding sequence of *E. faecalis* JCM 5803 (for 16S rRNA gene) or *S. suis* type strain NCTC10234^T^ (for *groEL*, *gyrB*, *sodA*, and *recN* genes) was used as an out-group to root the trees. The confidence values were obtained from 1000 replications. The bar represents sequence dissimilarity.

**Figure 2 pathogens-10-00834-f002:**
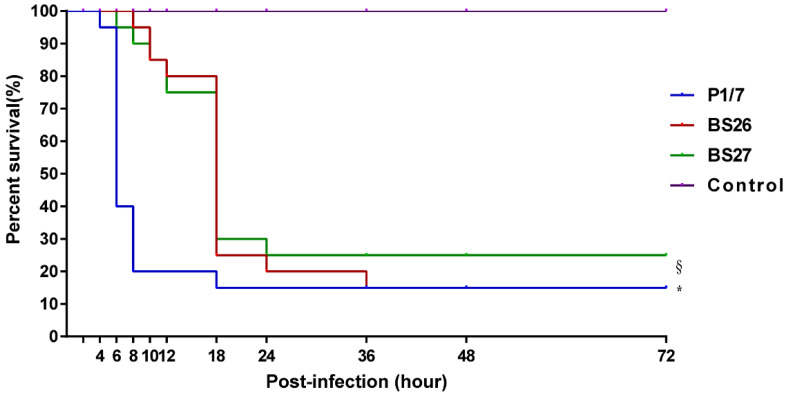
Survival curves of mice injected with 5 × 10^7^ CFU of live *S. parasuis* strain BS26, *S. parasuis* strain BS27, *S. suis* strain P1/7, and PBS only as control group. Overall survival rates of the two experiments for each group (calculated via the Kaplan-Meier method) at 2 h, 4 h, 6 h, 8 h, 10 h, 12 h, 18 h, 24 h, 36 h, 48 h, and 72 h post-infection were present in survival curve. The survival rates of different groups were compared using Log-rank test. *, significantly different (*p* < 0.05) compared to *S. parasuis* strains-infected group and control group. §, significantly different (*p* < 0.05) between *S. parasuis* strains-infected group and control group.

**Figure 3 pathogens-10-00834-f003:**
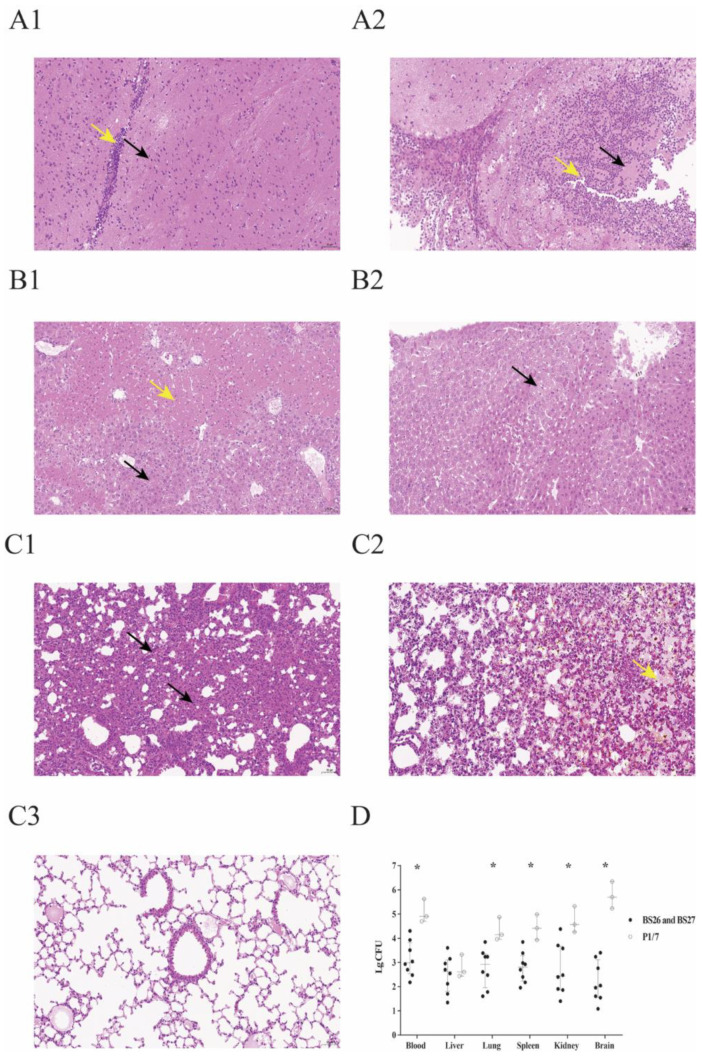
Histopathological changes and bacterial loads in the organs of survival mice in survival experiment. (**A1**) Micrograph of brain sample from a *S. parasuis* strain BS26-infected mouse at 72 h post-infection. Neuronal deformation (black arrowhead) and slight infiltration by lymphocytes (yellow arrowhead) were shown. (**A2**) Micrograph of brain sample from a mouse infected with *S. suis* strain P1/7 at 72 h post-infection. Neuronal necrolysis (black arrowhead) and significant infiltration by neutrophils (yellow arrowhead) were shown. (**B1**) Micrograph of liver sample from a mouse infected with *S. parasuis* strain BS26 at 72 h post-infection. Hepatocyte steatosis (black arrowhead) and coagulative necrosis (yellow arrowhead) were shown. (**B2**) Micrograph of liver sample from a mouse infected with *S. suis* strain P1/7 at 72 h post-infection. Edema of few hepatocytes was shown (black arrowhead). (**C1**) Micrograph of lung sample from a mouse infected with *S. parasuis* strain BS26 at 72 h post-infection. Alveoli wall thickening with diffused infiltration of neutrophils and lymphocytes (black arrowhead) was shown. (**C2**) Micrograph of lung sample from a mouse infected with *S. parasuis* strain BS26 at 72 h post-infection. Hemorrhagic foci (yellow arrowhead) are shown. (**C3**) Micrograph of lung sample from a mouse infected with *S. suis* strain P1/7 at 72 h post-infection. No significant histopathological lesions were observed. H&E staining, ×200 magnification, scale: 50 μm. (**D**) Bacterial loads in blood and organs of survival mice. Colonies were expressed as CFU/0.1 g for organ samples or CFU/mL for blood samples. Bacterial counts of individuals, including median with interquartile ranges, are presented. Significant difference in bacterial counts between *S. suis* and *S. parasuis*-infected groups was determined by Wilcoxon’s two-sample test. *, significantly different (*p* < 0.05) in bacterial counts between *S. suis* strain P1/7-infected group and *S. parasuis* strains-infected group.

**Figure 4 pathogens-10-00834-f004:**
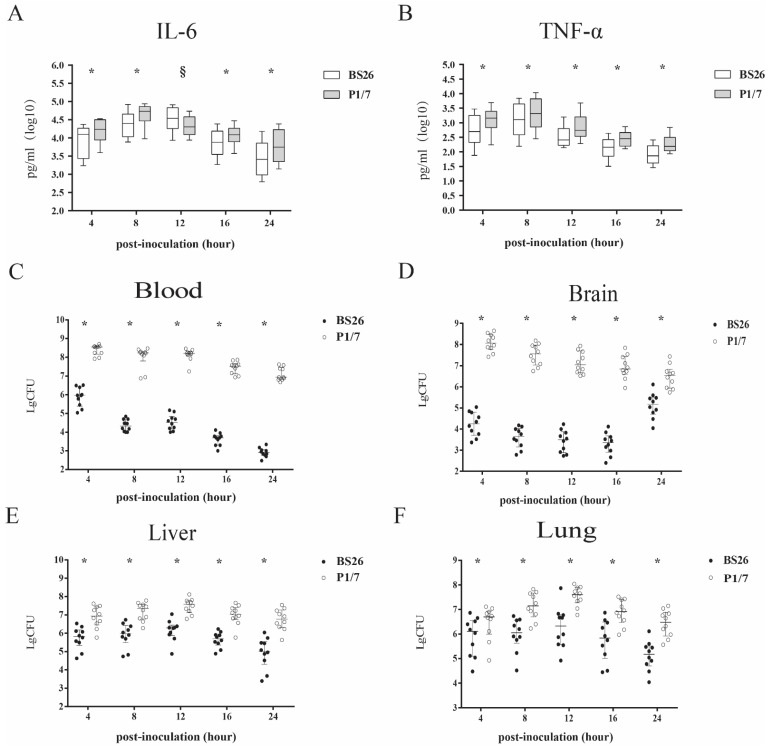
Production of pro-inflammatory cytokines IL-6 (**A**) and TNF-α (**B**) in sera and bacterial loads (**C**–**F**) of C57BL/6 mice infected with 1 × 10^7^ CFU of *S. suis* strain P1/7 and *S. parasuis* strain BS26. Median values with interquartile ranges of each group were used to express cytokine levels in sera. Bacterial counts of individuals, including median with interquartile ranges, were presented. Colonies were expressed as CFU/0.1 g for organ samples or CFU/mL for blood samples. Significant differences in bacterial counts and cytokine data between the two infected groups were determined by Wilcoxon’s two-sample test. * Cytokine levels and bacterial counts of mice infected with *S. suis* strain P1/7 were statistically higher than those of mice infected with *S. parasuis* strain BS26 (*p* < 0.05). § Cytokine levels of mice infected with *S. parasuis* strain BS26 were statistically higher than those of mice infected with *S. suis* strain P1/7 (*p* < 0.05).

**Figure 5 pathogens-10-00834-f005:**
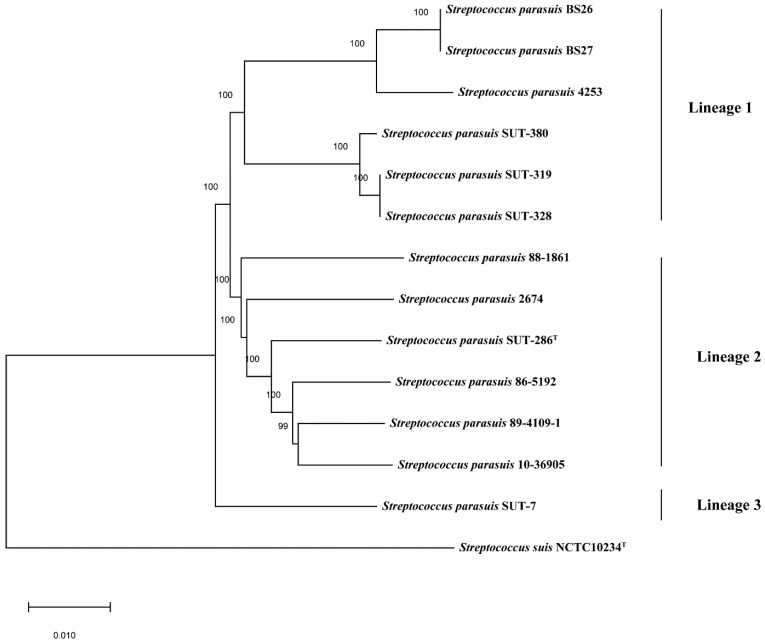
Core-genome phylogeny of 14 *Streptococcus* genomes in the present study. The tree was based on an alignment of 442 core genes of 13 *S. parasuis* genomes and 1 *S. suis* genome using the neighbor-joining method. The *S. suis* type strain NCTC10234^T^ was used as an out-group to root the tree. The bar represents sequence dissimilarity.

**Figure 6 pathogens-10-00834-f006:**
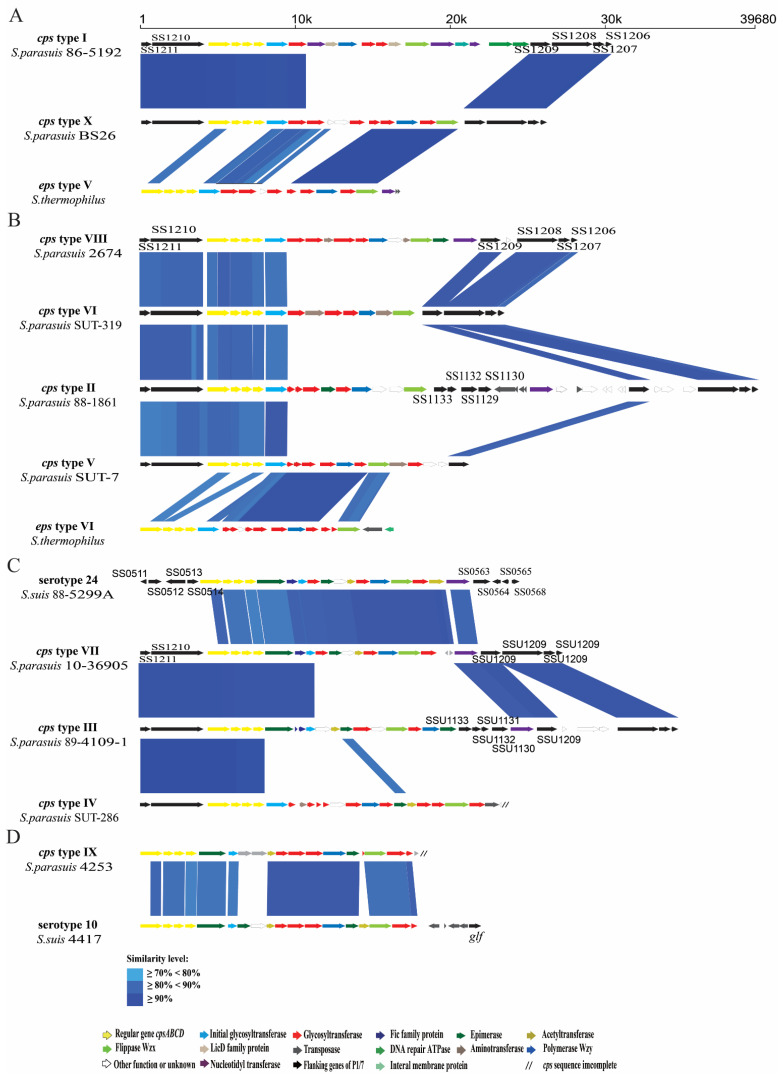
Schematic comparison of the *cps* gene loci among (**A**) *S. parasuis* type I, X, and *S. thermophilus* strain type V. (**B**) *S. parasuis* type II, V, VI, VIII, and *S. thermophilus* strain type VI. (**C**) *S. parasuis* type III, IV type VII, and *S. suis* serotype 24. (**D**) *S. parasuis* type IX, and *S. suis* serotype 10. Each colored arrow represents a gene whose predicted function is shown in the blown-up panel. The direction of the arrow indicates the direction of transcription. Regions of 70% identity were marked by blue shading.

**Table 1 pathogens-10-00834-t001:** The information on *S. parasuis* genomes used in the study.

Strains	Source	Location	Year	*cps* Type	Accession No.
BS26	Patient	China	2018	X	CP069079
BS27	Patient	China	2018	X	JAETXU000000000
86-5192	Diseased calf	United States	1980’	I	PRJNA171426
88-1861	Diseased pig	Canada	1980’	II	PRJNA171444
89-4109-1	Diseased pig	/	1980’	III	PRJNA171433
SUT-7	Healthy pig	Japan	/	V	DRX016751
SUT-286	Healthy pig	Japan	/	IV	DRX016752
SUT-319	Healthy pig	Japan	/	VI	DRX016753
SUT-328	Healthy pig	Japan	/	VI	DRX016754
SUT-380	Healthy pig	Japan	/	VI	DRX016755
10-36905	Healthy Bos taurus	United States	2010	VII	PRJNA590796
2674	Healthy pig	China	2014	VIII	POIG00000000
4253	Healthy cow	Switzerland	2018	IX	SHGT00000000

## Data Availability

The data presented in this study are openly available in the article and its Appendix A.

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
