# Peer review of "Investigation of the Genomic and Pathogenic Features of the Potentially Zoonotic Streptococcus parasuis"

_pathogens, 2021, doi:10.3390/pathogens10070834_

Round 1

Reviewer 1 Report

The manuscript by Jianping Wang et al. has been deeply revised and now is suitable for publication in Pathogens.

Author Response

No comments are needed to response

Reviewer 2 Report

Dear Authors,

I commend you for putting in the effort regarding the revision of the manuscript. I support the acceptance and the publication of the present version of the manuscript.

Author Response

No comments are needed to response

Reviewer 3 Report

It is confusing that the kind of data plotted (mean or median) is not always indicated in the figure captions. Please provide this information for each figure in the respective caption. Please indicate which statistical test was used to calculate significance levels for each graph. Error bars are not shown at all. Please include at least standard deviations in all relevant figures (Fig.s 2, 3D, 4A-D). Actually, in case of animal experiments it would be more informative to show data as individual value plots.

Fig. 3D: It is impossible to see the differences between the symbols, they are much to small.

Fig. 4. What about BS27? Were the respective data not collected for this strain?

Fig. 6: Letters in the figures are impossible to read.

The sequences referred to as being deposited under the accession numbers CP069079 and JAETXU000000000 could not be found in the GenBank database. Was the publication withheld at request of the authors?

Reviewer 4 Report

Dear Authors

It is interesting paper however my suggestion to improve this manuscript:

-line 481-485: according to what criteria was the breakpoints determined. Please provide the name of the organization

- authors give breakpoint for clindamycin, >1μg/mL. Could you check actually breakpoint? This brekapoints (CLSI), because it is >8mg/L. Againerythromycin, >1μg/mL? Could you give current brekapoints according to CLSI?

Round 2

Reviewer 3 Report

The authors addressed all concerns raised in my previous report.